**Data Availability Statement:** https://osf.io/x75jk/.

**Funding:** The author(s) received no specific funding for this work.

# Prevalence and determinants of using complementary and alternative medicine for the treatment of chronic illnesses: A multicenter study in Bangladesh

**Md. Shahjalal**[1,2], **Samar Kishor Chakma**[1,2], **Tanvir Ahmed**[1,2], **Irin Yasmin**[1,2], **Rashidul Alam Mahumud**[3,4], **Ahmed Hossain**[1,5]*

1 Department of Public Health, North South University, Dhaka, Bangladesh, 2 Research Rats, Dhaka, Bangladesh, 3 NHMRC Clinical Trials Centre, Faculty of Medicine and Health, The University of Sydney, Camperdown, New South Wales, Australia, 4 School of Business and Centre for Health Research, University of Southern Queensland, Toowoomba, Queensland, Australia, 5 Global Health Institute, North South University, Dhaka, Bangladesh

* ahmed.hossain@utoronto.ca

## Abstract

### Background

While conventional medicine (CM) is commonly used to treat non-communicable diseases (NCDs), complementary and alternative medicine (CAM) is gaining popularity as a health-care option in Bangladesh. We aimed to investigate the prevalence and factors associated with using CAM solely and using CAM in conjunction with CM for chronic illness treatment among NCD patients in Bangladesh.

### Methods

A multicenter cross-sectional study was conducted, including 549 adults with a confirmed chronic illness diagnosis from three tertiary care hospitals in Dhaka city. Interviews were used to gather socio-demographic data, while medical records were used to get information on chronic illnesses. A multinomial logistic regression model was used to determine the associated factors of utilizing CAM primarily and CAM use in conjunction with CM to manage the chronic disease.

### Results

Out of 549 NCD patients (282 women [51.4%], mean [standard deviation] age 45.4 [12.8] years), 180 (32.8%) ever used CAM for the treatment of chronic illness. Also, 15.3% of patients exclusively used CAM among the NCD patients, while 17.5% used CAM in conjunction with CM. Homeopathy medicine was the most prevalent type of treatment among CAM users (52.2%). Furthermore, 55.5% of CAM users said they used it due to its less adverse effects, and 41.6% trusted its effectiveness for chronic illness. Elderly patients (≥60 years) preferred CAM in complementary with CM, but they did not rely only on CAM. According to the multinomial regression analysis, unmarried patients, predominantly in the younger age

**Competing interests:** The authors have declared that no competing interests exist.

group, adopted CAM significantly for chronic illness treatment (Relative risk ratio, RRR = 0.29, 95% CI = 0.12–0.71, reference = Unmarried). Patients in the high-income group used CAM in conjunction with CM (RRR = 6.26, 95% CI = 1.35–18.90, reference: low-income), whereas patients in the high-income group did not rely on CAM alone (RRR = 0.99, 95% CI = 0.34–2.85).

## Conclusion

Although CM remains the mainstream of health care in Bangladesh, CAM services play an essential role in people's health care, particularly in treating chronic illnesses. Physicians of Bangladesh should be aware that their patients may be using other services and be prepared to ask and answer questions regarding the risks and benefits of using CAM in addition to regular medical care. Thus, clinicians required to follow best-practice guidelines, which are currently not practiced in Bangladesh, when disseminating information regarding integrative techniques that combine CM and CAM approaches.

## Introduction

Complementary and alternative medicines (CAMs) are a non-mainstream approach that do not fit under the umbrella of traditional medicine. Acupuncture, homeopathy, aromatherapy, meditation, and colonic irrigation are examples of these medications and treatments. For millennia, CAM has been a popular way to meet people's basic healthcare needs [1]. CAM is the primary source of treatment for millions of individuals. In other cases, it is the only source of treatment due to a lack of adequate healthcare access, cultural differences, and healthcare costs [2]. Although conventional medicine (CM) has improved in recent decades, the use of CAM for illness prevention, control, and management has expanded around the world [2, 3].

A considerable proportion of people in developed countries uses CAM healthcare services [4–6], which is equally true in developing countries [7–9]. Overall, the prevalence of using CAM ranged from 20% to 97.4% in South-East Asia [10]. More than 70% of the population in developing countries still depends on CAM treatment despite conventional medicine's progress [7]. According to a recent survey, 35.2% of diabetes patients in Bangladesh used CAM for their diabetic management [11].

Chronic diseases are caused by genetic, physiological, environmental, and behavioral variables, also known as non-communicable diseases (NCDs) [12]. NCDs place a considerable strain on the healthcare system in Southeast Asia, contributing to the high prevalence of early death (<70 years) [13]. Low- and middle-income countries (LMICs), such as Bangladesh, account for approximately 77% of overall NCD fatalities [12].

Moreover, NCDs are a significant source of morbidity and mortality in Bangladesh, accounting for 61% of all fatalities [14]. The most common NCDs include cardiovascular diseases, diabetes mellitus, cancer, and chronic respiratory diseases. As the proportions of these diseases increase year after year, these illnesses impose a significant burden on healthcare facilities [15]. Due to the increasing burden of NCDs, new challenges are arising around its long-term management [16]. Despite the enormous advancement of mainstream medicine, the usage of CAM continues to grow across the country [17]. Although CAM is not considered a part of mainstream health care systems, it has been employed in Bangladesh as an alternative to traditional treatment [17].

Patients seeking medical attention in Bangladesh are diverse, and disparity can be linked to physician bias or socioeconomic status and patients' understanding of risks and benefits, and healthcare system barriers [16]. Different CAM techniques, such as Ayurvedic, Unani, Homeopathy, Naturopathy, and other folk practitioners, are commonly used in Bangladesh to address medical needs [17]. Bangladesh's geographical location and climate conditions are conducive to the growth and usage of CAM. Despite the widespread acceptance of CAM alongside traditional treatment, there is still a lack of data on the prevalence, utilization, and associated factors of using CAM for chronic illness among NCD patients in Bangladesh. This study will look at the following goals: (i) Compute the prevalence of using CAM to manage chronic illness and categorizing patients by demographic subgroups, ii) determine the associated factors of exclusively CAM use for chronic disease, and iii) determine the factors associated with uses of CAM in conjunction with CM to manage chronic illness.

## Materials and methods

### Study design and settings

The study used a cross-sectional survey approach and was conducted among patients with NCDs who received healthcare services from the three tertiary care hospitals in Dhaka. The three hospitals were chosen conveniently, and the survey took place from January 12 to March 7, 2021. The following inclusion and exclusion criteria were set before data collection.

**Inclusion criteria:**

1. All of the participants had a chronic condition and had been taking medication for at least six months.

2. The chronic illnesses included: diabetes, hypertension, cardiovascular diseases (Coronary artery disease, Heart attack, Heart failure, Strokes or Rheumatic Heart Disease), chronic respiratory illness (Chronic obstructive pulmonary disease or asthma), chronic musculoskeletal disorders (chronic low back pain), rheumatic diseases, kidney diseases, and cancer.

3. The chronic condition of the participants was diagnosed using health care facilities such as hospitals, clinics, or CAM facilities, as well as by consultation with an expert clinical practitioner or CAM practitioner.

4. Age: 20–80 years.

5. The participants agreed to the consent form to respond to CAM uses and completed the interview.

**Exclusion criteria:**

1. We excluded pregnant women from the study to reduce scientific complexity, as there is a chance that a patient seeking medical attention for pregnancy rather than chronic illness would be included.

2. Due to a lack of resources for immediate diagnosis of chronic or acute pain illness, individuals who reported pain for neuropathy or muscular pain were excluded.

3. Patients who reported self-medication for a chronic illness were excluded as these patients do not engage in health-care consultations.

4. Patients with psychiatric disorders were ruled out by inquiring if they took any medicine for psychiatric symptoms or illegal drugs.

## Sample size

The sample size for the survey was calculated using a 20% anticipated prevalence of CAM use for NCDs, with a 95% level of confidence and a 5% margin of error. The sample size was estimated using the epiR package of the R program 3.6.1. Bennett, Woods, and Smith described the formula for estimating sample size in a cluster sampling procedure (1991). The prevalence estimate is used to calculate the sample size required for the survey, resulting in the largest sample size possible. After adjustments, a total of at least 540 patients were included in the study. For the survey, at least 200 patients per health facility were used after adjustments.

## Sample selection

In the first stage, three tertiary care hospitals from Dhaka city were selected based on the study conveniences. Islami Bank Hospital, Dhaka Medical College Hospital, and BIRDEM were the three hospitals. Patients were chosen by field-enumerators in the second stage by a systematic sampling procedure. When patients arrived at the specified hospitals' NCD departments, they were assessed and asked about their diagnosis of chronic illness and duration of suffering for chronic disease. Then, using an alternative participant selection in the lines of visiting patients, at least 200 patients were selected from each hospital during the study period. At least five patients were interviewed each day during hospital working hours with the assistance of hospital physicians. A team of data collection enumerators interviewed the selected patients, and the hospital physician confirmed the patient's NCD status. If the patient did not consent to data collection, they were not interviewed. Finally, data from 549 patients were gathered for analysis, and the flowchart of data collection is given in Fig 1.

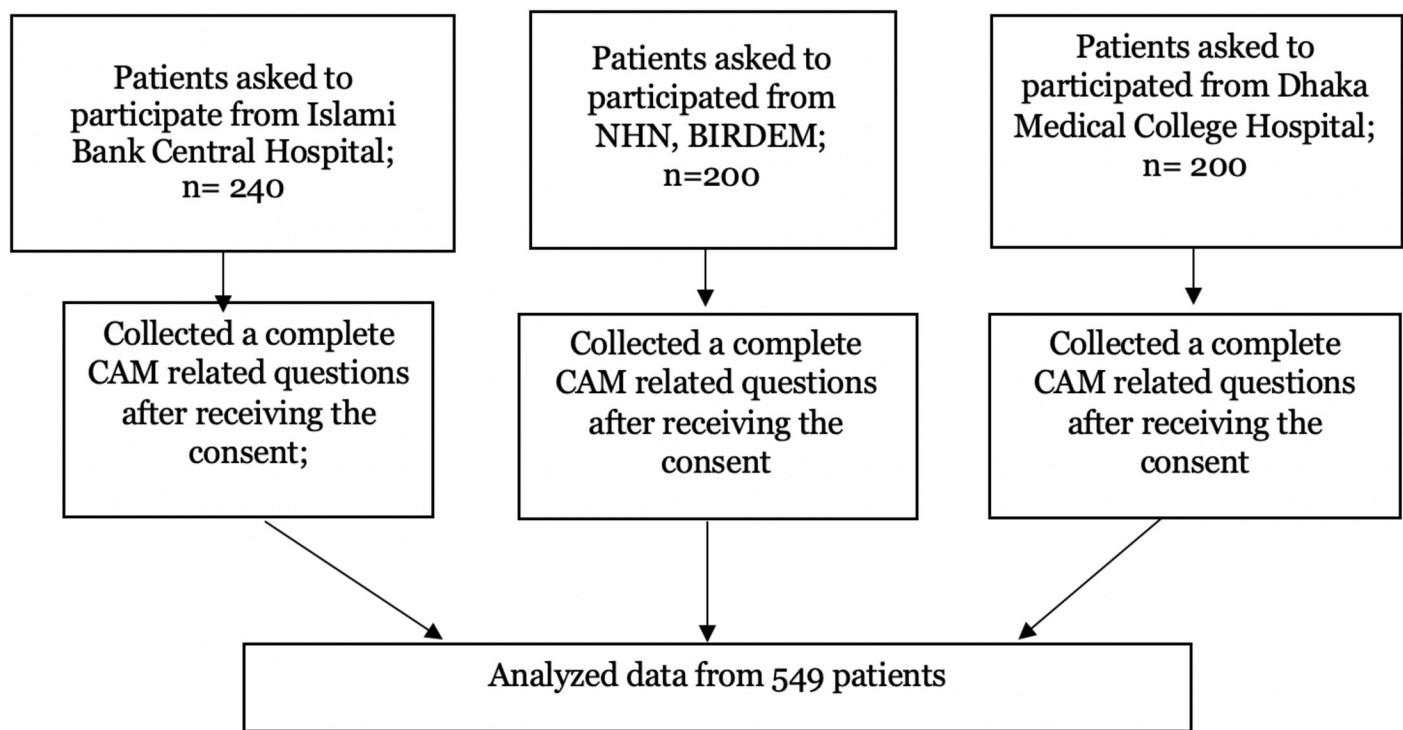

**Fig 1. Flow chart of participants inclusion for analysis.**

## Measures

**Independent variables.** Independent variables included patients' sociodemographic characteristics and healthcare data on chronic illness. Gender, age group (20–39, 40–59, 60+ years), marital status (married, never married), years of schooling (1–5 years, 6–12 years, 12+ years of education), location of residence based on urbanicity (urban, rural), and employment status were among the sociodemographic characteristics collected in the study. Their monthly household income measured the respondents' economic situation. Monthly household income was divided into three categories: less than 20000 BDT (about US$ 238), 21000–50000 BDT, and >50000 BDT. We used two questions to determine the level of knowledge about CAM for chronic illness. The first is "if the patients have heard of CAM for chronic illness treatment", and the second is "whether they are aware of CAM's effectiveness in chronic illness management".

**Outcome variable.** The survey's outcome variable was the use of complementary and alternative medicine (CAM). The mode of treatments in CAM definition was included as following: Ayurveda (Ayurvedic medicine), herbalism, homeopathy, Unani medicine, and traditional Chinese medicine. Acupuncture, aromatherapy, massage therapy, meditation, and spiritual healing were not included in the study definition of CAM because they do not contain medication for treating chronic illness patients. The study had three options for defining CAM: (i) Use CM exclusively for NCDs, (ii) Use CAM exclusively for NCDs, and (iii) Use CAM in conjunction with CM for NCDs. The outcome was determined using two questions. "Did you ever use complementary and alternative medicine (Ayurveda, herbalism, homeopathy, Unani medicine, and traditional Chinese medicine) for NCDs?" is the first question. The second question is, "Have you ever combined CAM and CM for NCDs?" The replies to each question were separated into two categories and recorded as yes or no.

## Ethical permission

The participants gave verbal agreement for the study because they intended not to sign or leave fingerprints on any paper, and many of them were analphabetic. The respondents were informed that all of the information obtained would be kept confidential and utilized solely for research purposes. They were, however, given a consent form that included specific contact information for the research investigators in case they had any questions in the future. The study protocol was approved by the Institutional Review Board of North South University in Bangladesh (2020/OR-NSU/IRB/1006).

## Statistical analysis

Data were analyzed using R 3.6.2. The questionnaire, R scripts, and data are available at https://osf.io/x75jk/. The CAM usage exclusively and CAM usage in conjunction with CM among the respondents were presented using descriptive statistics (frequencies, percentage). Therefore, a cross-tabulation was performed between the outcome measure (CM usage, CAM usage, and both CM and CAM usage) and the covariates. A multinomial regression model was fitted to examine the association of socio-demographic variables and CAM usage against chronic illness. The multinomial regression coefficient from the model was exponentiated and presented as relative risk ratios (RRR) along with corresponding 95% confidence intervals (CI). Here, RRR is defined as the ratio of the probability of an outcome in the exposed group to the probability of an outcome in the unexposed group [18]. As some readers may find odds ratios (OR) easier to interpret than RRR, the study included two binary logistic regressions examining, i) between CM usage exclusively and CAM usage exclusively, and ii) between CM usage exclusively and CAM usage in conjunction with CM in S3 and S4 Figs, respectively. The

study also obtained variance inflation factors (VIF) in the logistic regression models to evaluate potential multicollinearity.

## Results

### Participants' characteristics

The mean ± SD age of the 549 participants was 45.4 ± 12.8 years. A total of 549 participants were interviewed, and their sociodemographic characteristics are given in Table 1. More than half of the participants (51.4%) were female. The majority of participants' age was between 40 years and 59 years (47.7%). Most of them had none or 1–5 years of schooling (42.6%). Besides, most of the participants were married (91.4%). The majority of participants were living in the urban area (81.4%). Most of the survey respondents were housewives or not working (65.6%), and their monthly household income was 20001–50000 BDT (71.9%). Moreover, patients' most common non-communicable diseases were: high blood pressure (35.2%) and diabetes (33.3%). In S1 Fig, the association between age, gender, education level, and marital status was discussed. The majority of the female participants appeared to have completed less than five years of schooling. Females had lower levels of education than males. In addition, the majority of the unmarried population was male and younger in age.

**Table 1. Distribution of participants according to CAM and CM users.**

| Participants' characteristics | CM Only (n = 369, 67.2%) | CAM Only (n = 84, 15.3%) | CAM in conjunction with CM (n = 96, 17.5%) | Total (n = 549) |
|---|---|---|---|---|
| **Gender** | | | | |
| Male | 179 (67.0%) | 37 (13.9%) | 51 (19.1%) | 267 (48.6%) |
| Female | 190 (67.4%) | 47 (16.7%) | 45 (16.0%) | 282 (51.4%) |
| **Age (years)** | | | | |
| 20–39 | 132 (68.8%) | 30 (15.6%) | 30 (15.6%) | 192 (35.0%) |
| 40–59 | 173 (66.0%) | 41 (15.6%) | 48 (18.3%) | 262 (47.7%) |
| ≥60 | 64 (67.4%) | 13 (13.7%) | 18 (18.9%) | 95 (17.3%) |
| **Schooling** | | | | |
| <= 5 years | 158 (67.5%) | 41 (17.5%) | 35 (15.0%) | 234 (42.6%) |
| 6–12 years | 137 (65.2%) | 36 (17.1%) | 37 (17.6%) | 210 (38.3%) |
| 12+ years | 74 (70.5%) | 7 (6.7%) | 24 (22.9%) | 105 (19.1%) |
| **Marital status** | | | | |
| Married | 346 (68.9%) | 71 (14.1%) | 85 (16.9%) | 502 (91.4%) |
| Never married | 23 (48.9%) | 13 (27.7%) | 11 (23.4%) | 47 (8.6%) |
| **Location of residence** | | | | |
| Urban | 303 (67.8%) | 70 (15.7%) | 74 (16.6%) | 447 (81.4%) |
| Rural | 66 (64.7%) | 14 (13.7%) | 22 (21.6%) | 102 (18.6%) |
| **Employment status** | | | | |
| No employment | 239 (66.4%) | 65 (18.1%) | 56 (15.6%) | 360 (65.6%) |
| Had an employment | 130 (68.8%) | 19 (10.1%) | 40 (21.2%) | 189 (34.4%) |
| **Monthly household income (BDT)** | | | | |
| ≤20000 | 40 (80.0%) | 8 (16.0%) | 2 (4.0%) | 50 (9.1%) |
| 20001–50000 | 259 (65.6%) | 66 (16.7%) | 70 (17.7%) | 395 (71.9%) |
| 50000+ | 70 (67.3%) | 10 (9.6%) | 24 (23.1%) | 104 (18.9%) |
| **Knowledge on CAM** | | | | |
| No | 108 (96.3%) | 0 (0%) | 4 (3.7%) | 108 (19.7%) |
| Yes | 265 (65.6%) | 84 (19.0%) | 92 (20.9%) | 441 (80.3%) |

## Prevalence of CAM use to manage chronic illness

Table 1 displays the prevalence of complementary and alternative medicine (CAM) use for chronic illness by sociodemographic categories. It appears that 180 of the 549 respondents (32.8%) utilized CAM for their chronic illness. Among the 180 CAM users, Homeopathy was utilized by 94 (52.2%), Ayurveda by 48 (26.6%), Unani by 29 (16%), and other CAM by 9 (5.2%) patients. S2 Fig displays a pie chart depicting the various types of CAM used by the patients. Furthermore, 84 (15.3 percent) of patients used CAM exclusively for sickness, whereas 94 (17.5 percent) utilized both CAM and CM at the same time for the management of chronic conditions. The remaining 369 (67.2%) of the 549 patients relied solely on CM to manage their chronic illnesses.

According to the findings, 16.7% of 282 females and 13.9 percent of 267 males utilized CAM exclusively. Only CAM use was similar among persons in the 20–59 year age group, while it was low (13.7%) among the elderly (60+) age group. Both CM and CAM are used more frequently as people get older. Furthermore, among patients aged 20 to 39, the use of solely CM was common (68.8%).

Between respondents with 12+ years of schooling and those with less education, there was a notable change in the percentage of CAM use. Respondents with more than 12 years of schooling were less likely to use complementary and alternative medicine (6.7 percent). Respondents with a greater level of education, on the other hand, are more likely to use both CM and CAM to treat their sickness.

When compared to married respondents, a higher number of never-married respondents utilized CAM solely (27.7% for married versus 14.1% for never-married). Also, the percentage of never-married respondents who used both CAM and CM was greater. Furthermore, when compared to rural inhabitants, urban residents reported a higher percentage of CAM use (15.7% for urban versus 13.7% for rural). Rural residents, on the other hand, utilized a higher percentage of CAM usage in conjunction with CM than city dwellers (21.6% versus 16.6%).

Employed respondents used CAM at a lower rate than unemployed respondents. Only 10.1% of the 189 working respondents stated they used complementary and alternative medicine (CAM), compared to 18.1% of the 360 jobless respondents. However, employed respondents had a larger rate of both CM and CAM users than unemployed respondents (15.6%). Those with a monthly family income of more than 50000 BDT were more likely to employ both CM and CAM to treat their sickness. According to the findings, 108 (19.7%) of 549 patients did not know about CAM and had never utilized it to treat their ailment. On the other hand, 441 (80.3%) of the 549 patients knew about CAM, and around 40% of them had utilized it for their chronic illness.

## Distribution of CAM use by clinical features

Table 2 shows the distribution of CAM use and CM use for chronic illness according to the clinical characteristics of the patients. The majority of the participants visited hospitals with hypertension (35.2%) and diabetes problems (33.3%). Also, 78.8% of patients with hypertension were more likely to use CM alone, while 61.7% of diabetic patients were more likely to use CM alone. Diabetic individuals were also more likely to employ both CAM and CM to treat their ailment. Patients with respiratory sickness and musculoskeletal disorders were shown to be heavily reliant on CAM. These two categories of patients used CAM and CM for their illnesses at a higher rate than the other groups of diseases.

**Table 2. The percentage of the use of CAM by clinical features of the patients.**

| Clinical features | CM Only (n = 369, 67.2%) | CAM Only (n = 84, 15.3%) | CAM in conjunction with CM (n = 96, 17.5%) | Total (n = 549) |
|---|---|---|---|---|
| High blood pressure/ Hypertension | 152 (78.8%) | 33 (17.1%) | 8 (4.1%) | 193 (35.2%) |
| Diabetes | 113 (61.7%) | 21 (11.5%) | 49 (26.8%) | 183 (33.3%) |
| CVD | 21 (65.6%) | 8 (25.0%) | 3 (9.4%) | 32(5.8%) |
| Respiratory illness | 9 (40.9%) | 6 (27.3%) | 7 (31.8%) | 22 (4%) |
| Musculoskeletal disorder | 13 (38.2%) | 7 (20.6%) | 14 (41.2%) | 34 (6.2%) |
| Others (e.g., Cancer, CKD, Rheumatic diseases) | 61 (71.8%) | 9 (10.6%) | 15 (17.6%) | 85 (15.5%) |

## Multinomial logistic regression model: Determinants of the utilization of CAM to manage NCDs

Table 3 shows the results of a multinomial logistic regression model for predicting the usage of CAM as well as the use of CAM in conjunction with CM. When compared to unmarried patients, married patients were 71% less likely to use exclusively complementary and alternative medicine (RRR = 0.29, 95% CI = 0.12–0.71). Similarly, married persons were 62% less likely to use both CAM and CM than unmarried people (RRR = 0.38, 95% CI = 0.16–0.90). Furthermore, patients with a monthly household income of more than 50000 BDT were 6.26 times more likely than low-income families to utilize both CAM and CM (RRR = 6.26, 95% CI = 1.35–28.90). Patients' gender, age, education, locality, and employability, on the other hand, were unrelated to their use of CAM.

## Reasons for CAM use to manage chronic illness

Patients who had previously used complementary and alternative medicine for their disease were questioned about their motives for doing so, and the results are shown in Fig 2. For such a question, multiple responses were collected. According to the findings, 55.5% of the 180

**Table 3. Multinomial logistic regression model to understand associated factors on the use of CAM.**

| Participants' characteristics | CAM only | | CAM in conjunction with CM | |
|---|---|---|---|---|
| | RRR | 95% CI | RRR | 95% CI |
| **Gender (reference: female)** | | | | |
| Male | 0.96 | 0.54–1.72 | 1.17 | 0.68–2.02 |
| **Age group (reference: 20–39 years)** | | | | |
| 40–59 years | 1.34 | 0.72–2.50 | 1.66 | 0.91–3.01 |
| ≥60 years | 1.02 | 0.44–2.37 | 1.93 | 0.88–4.20 |
| **Marital status (reference: never married)** | | | | |
| **Married** | **0.29** | **0.12–0.71** | **0.38** | **0.16–0.90** |
| **Schooling (reference: <= 5 years)** | | | | |
| 6–12 years | 0.96 | 0.56–1.67 | 1.15 | 0.66–2.01 |
| 12+ years | 0.40 | 0.15–1.05 | 1.21 | 0.71–2.29 |
| **Location of residence (reference: urban)** | | | | |
| Rural | 0.99 | 0.52–1.90 | 1.36 | 0.77–2.40 |
| **Employment status (reference: Housewife or not employed)** | | | | |
| Employed | 0.61 | 0.32–1.17 | 1.50 | 0.86–2.62 |
| **Monthly family income (reference: < = 20000 BDT)** | | | | |
| 20001–50000 BDT | 1.30 | 0.57–2.96 | **5.39** | **1.26–13.06** |
| 50000+ BDT | 0.99 | 0.34–2.85 | **6.26** | **1.35–18.90** |

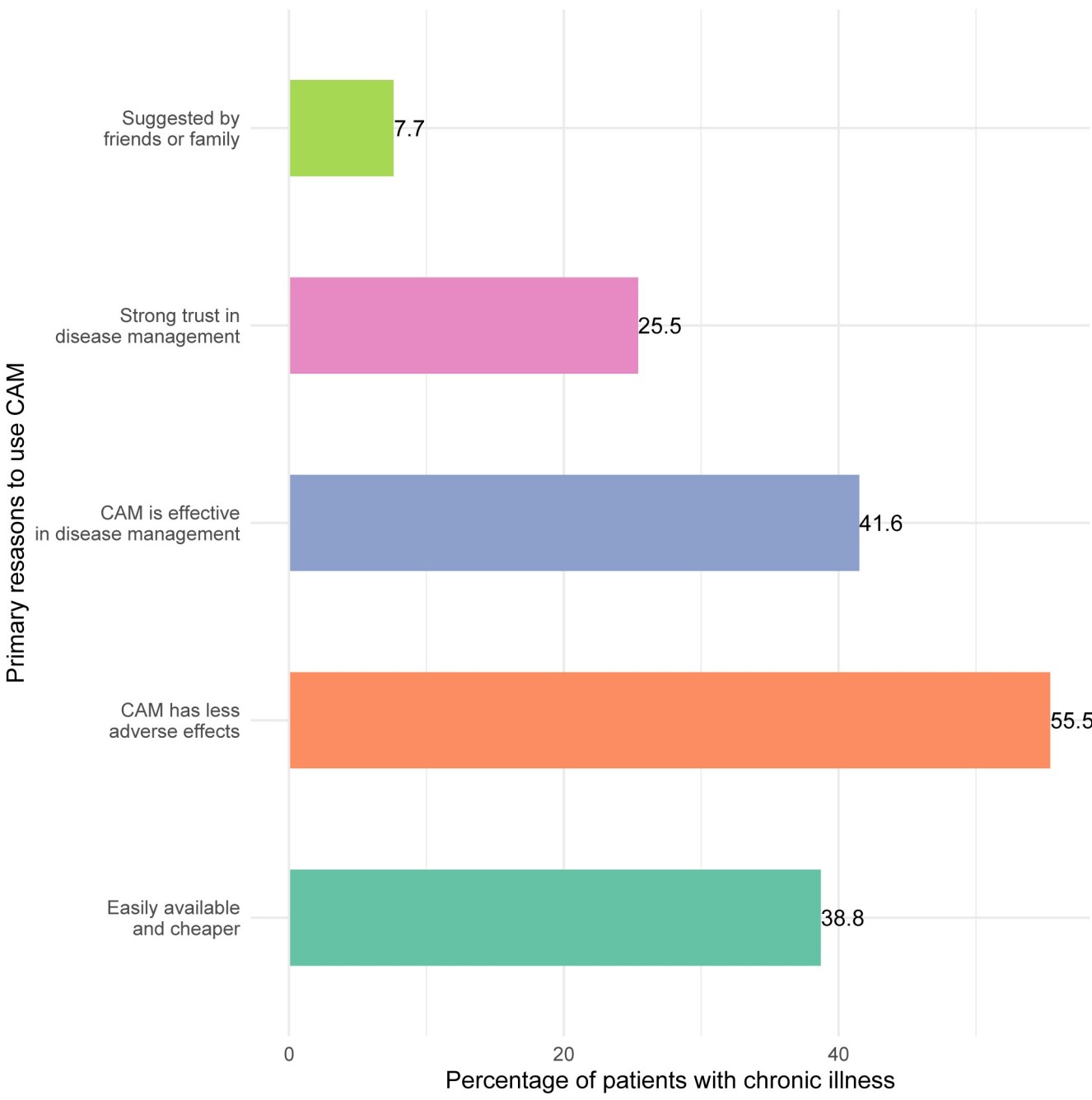

**Fig 2. Reasons for CAM use among patients.**

CAM users said they used it to avoid adverse side effects. Furthermore, 41.6% of the CAM users stated it was effective in the management of chronic illnesses, while 38.8% said it was easier to obtain from local services and less expensive.

## Discussion

This study investigates the prevalence and pattern of complementary and alternative medicine (CAM) usage and the factors that influence CAM use among Bangladeshi adult patients with

NCDs. In this study, one-third of the participants had utilized CAM to manage chronic diseases at some point in their lives. This result is similar to what was previously reported among diabetic patients in Bangladesh (35.2%) [11]. A study in Malaysia found that 63.9% of patients with chronic conditions used CAM, and another study in North-India found the rate was 53.2%, which is significantly higher than our estimate [19, 20]. Besides, Singapore (22.7%) and South Africa (27.2%) had lower rates of CAM use among patients with chronic illness [21, 22]. Moreover, the most prevalent reasons for CAM use were a few side effects, improved efficacy, and that CAM is less expensive and more readily available. These causes are comparable to the findings of earlier investigations [11, 17–21].

According to this study, patients with NCDs exhibited a significant intention to use CAM in conjunction with CM (17.5%) to manage their chronic diseases. A survey in the United States indicated that 54.9% of individuals used CAM in combination with CM [23], while another study in North-India found that 49.8% of patients used CAM in conjunction with CM [20]. This finding raises concerns regarding drug interactions between these two types of treatment methods. To establish the efficacy and safety of combined treatment, it appears necessary to look into the scientific data.

According to the findings of this survey, socio-demographic characteristics such as marital status and a high monthly household income were significantly related to CAM use in conjunction with CM among NCD patients, which is similar to earlier studies from Malaysia, Nepal, and Pakistan [9, 19, 24]. These findings could point to a trend among CAM users for high-income people to seek every available alternative for their health care benefit and well-being. However, a recent study found that, in addition to the four income levels, both the lowest and highest socioeconomic groups in China demonstrated an increased preference for CAM [25].

The majority of females used CAM solely when they were younger, while most males preferred to use CAM in combination with CM. A study in the United States found that females were more likely to use CAM than males [23]. On the other hand, a survey conducted in North India and another study conducted in Bangladesh with diabetic patients observed no gender differences in CAM use [11, 2o].

The global epidemic of chronic diseases is driven by population aging, yet there is significant untapped potential to change the relationship between chronological age and health [26]. As a result, obtaining medical attention at the onset of sickness is critical for managing a chronic condition. Our results suggest the percentage of CAM use in combination with CM was high among the older (≥60) age group patients. This result is consistent with the other studies in South-East Asian countries like India, Malaysia, and China [19, 22, 25]. One possible explanation is that younger people seek alternative and low-cost remedies for mild issues, while elderly patients seek traditional medical aid for severe complications.

According to this data, patients with a high level of education were more likely to use combination CAM and CM. Patients with an insufficient level of education, on the other hand, were more likely to use CAM solely. This finding backs with a previous study conducted in Bangladesh, South Africa, and India, which found that primary school students were more likely to use CAM than those with a higher degree [11, 20, 21]. Another study in Canada saw an increased level of education increases the use of CAM in conjunction with CM [27]. Lack of knowledge was substantially linked to lower education and income levels as a factor for non-use of CAM.

## Strengths and limitations

This research has several positive aspects. First, we gathered data with the cooperation of university graduate enumerators and physicians from the surveyed hospitals. By involving

physicians from hospitals, we were able to reduce information bias. Second, using a systematic sampling strategy to find patients within a hospital helped reducing selection bias. Third, asking question if CAM was used in combination with CM to determine CAM usage, which helped categorizing CAM users in three groups, which was missing in most literature. Several limitations should be taken into account when interpreting the results. First, this study could not capture the trend of CAM usage over time, as we considered a cross-sectional study. Second, the study did not view the availability of CAM services. This could be significant because the prevalence of CAM usage among respondents may be mitigated by accessing or using available health services. Also, the study did not consider other variables such as duration of chronic diseases, family history of treatment methods, access to health services, etc. These variables could have confounding effects on the prevalence of using CAM or using CAM in combination with CM.

## Conclusion

Although patients with chronic diseases utilized a high percentage of CM, CAM is also widely used for chronic illness treatment in Bangladesh. A large proportion of patients viewed the use of CAM in conjunction with CM as complementary rather than alternative. As a result, physicians in Bangladesh should be aware that their patients may be utilizing other services and be ready to ask and answer questions about the risks and advantages of using CAM in addition to traditional medical care. This study emphasize clinicians' requirement to follow a best-practice guideline when disseminating information regarding integrative techniques that combine CM and CAM approaches to manage chronic illness.

## Supporting information

**S1 Fig. Types of CAM utilized by the patients.**
(DOCX)

**S2 Fig. Relationship between age, gender, education level and marital status.**
(DOCX)

**S3 Fig. Multivariable logistic regression model with utilization of CAM exclusively.**
(DOCX)

**S4 Fig. Multivariable logistic regression model with utilization of CAM in combination with CM.**
(DOCX)

## Acknowledgments

All authors acknowledge the graduate students from North South University who were involved in data collection. We are also thankful to the hospital authorities for their approval and support during data collection. We would also like to thank the reviewers for insightful comments that improved the presentation and clarity of our manuscript.

## Author Contributions

**Conceptualization:** Md. Shahjalal, Samar Kishor Chakma, Tanvir Ahmed, Irin Yasmin, Rashidul Alam Mahumud, Ahmed Hossain.

**Data curation:** Md. Shahjalal, Tanvir Ahmed, Irin Yasmin, Ahmed Hossain.

**Formal analysis:** Ahmed Hossain.

**Investigation:** Md. Shahjalal, Rashidul Alam Mahumud.

**Methodology:** Ahmed Hossain.

**Resources:** Samar Kishor Chakma.

**Supervision:** Ahmed Hossain.

**Writing – original draft:** Md. Shahjalal, Ahmed Hossain.

**Writing – review & editing:** Samar Kishor Chakma, Tanvir Ahmed, Irin Yasmin, Rashidul Alam Mahumud, Ahmed Hossain.

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
