## [Decision Letter · Decision Letter 0]

10 Dec 2021

PONE-D-21-26170Prevalence and determinants of using complementary and alternative medicine for the treatment of chronic illness: A multicenter study in BangladeshPLOS ONE

Dear Dr. Hossain,

Thank you for submitting your manuscript to PLOS ONE. After careful consideration, we feel that it has merit but does not fully meet PLOS ONE’s publication criteria as it currently stands. Therefore, we invite you to submit a revised version of the manuscript that addresses the points raised during the review process.

We look forward to receiving your revised manuscript.

Kind regards,

Bashir Alsiddig Yousef, Ph.D

Academic Editor

PLOS ONE

Journal Requirements:

2. Please provide additional details regarding participant consent. In the ethics statement in the Methods and online submission information, please ensure that you have specified : 1) whether the ethics committee approved the verbal/oral consent procedure, 2) why written consent could not be obtained, and 3) how verbal/oral consent was recorded. If your study included minors, please state whether you obtained consent from parents or guardians in these cases. If the need for consent was waived by the ethics committee, please include this information.

Reviewers' comments:

Reviewer's Responses to Questions

**Comments to the Author**

1. Is the manuscript technically sound, and do the data support the conclusions?

Reviewer #1: Yes

Reviewer #2: Yes

2. Has the statistical analysis been performed appropriately and rigorously? 

Reviewer #1: I Don't Know

Reviewer #2: Yes

3. Have the authors made all data underlying the findings in their manuscript fully available?

Reviewer #1: Yes

Reviewer #2: Yes

4. Is the manuscript presented in an intelligible fashion and written in standard English?

Reviewer #1: Yes

Reviewer #2: Yes

5. Review Comments to the Author

Reviewer #1: - In introduction, add a concise definition for CAM

-In the sampling (page 5), hospital selection based on convivence, which may decrease the external validity if the study

- In methodology, Study pupation, (page 5) pregnant excluded, Why?, Aslo individuals using self medications were excluded, why?.

-

Reviewer #2: Attached comments for the author:

Minor change in the tiltle " illnesses".

Minor comments in abstract.

Materials should be clear title.

Results : should be explain with more clear figure.

The type of referencing should be consistent.

6. PLOS authors have the option to publish the peer review history of their article (what does this mean?). If published, this will include your full peer review and any attached files.

Reviewer #1: No

Reviewer #2: **Yes: **Abdelgadir Alamin Abdelgadir

---

## [Author Response · Author response to Decision Letter 0]

18 Dec 2021

December 18, 2021

Bashir Alsiddig Yousef, Ph.D

Academic Editor

PLOS ONE

PONE-D-21-26170 

Prevalence and determinants of using complementary and alternative

medicine for the treatment of chronic illnesses: A multicenter study in Bangladesh

Dear Professor Bashir Alsiddig Yousef,

Thank you very much for your editorial suggestions and the reviewers’ comments. They were accommodating. Please find enclosed an itemized list of responses along with the revised manuscript. 

In our response to the reviewer, we used regular font for the comments/questions by the referees and regular, bold font for our responses, which are shown immediately following the questions/comments.

Thank you once again for the opportunity to submit a revised manuscript.

Ahmed Hossain, PhD

Professor, Department of Public Health

Director, Global Health Institute.

North South University.

Reviewer #1:

 - In introduction, add a concise definition for CAM

Authors: Thank you very much for your insightful comment. We included the following definition in the introduction: 

Complementary and alternative medicines (CAMs) are a non-mainstream approach that do not fit under the umbrella of traditional medicine. Acupuncture, homeopathy, aromatherapy, meditation, and colonic irrigation are examples of these medications and treatments.

-In the sampling (page 5), hospital selection based on convivence, which may decrease the external validity in the study.

Authors: Thank you very much for your wonderful comment. We agree with you that the convenient sampling has its own limitations of taking under- or over-representation sample of the population. It is the most commonly used sampling technique in public health as it is incredibly prompt, uncomplicated, and economical. To reduce the biasness in our study we choose three popular public hospitals in Dhaka. We also reduced bias in sampling by using systematic sampling in patients’ selection. Thus, we used both convenience sampling and probability sampling techniques to draw a more accurate result. The probability aspect used, along with convenience sampling, helped us to reduce bias in the results. 

- In methodology, Study pupation, (page 5) pregnant excluded, Why?

Authors: We excluded pregnant women from the study to reduce scientific complexity, as there is a chance that a patient seeking medical attention for pregnancy rather than chronic illness would be included. Pregnant women are scientifically complex, owing to their physiologic and ethical complexity. Also, pregnant women with substantial medical issues (e.g., diabetes, inflammatory bowel disease, depression, and epilepsy) may have a false positive reaction as a chronic illness patient. Because these factors could have a negative impact on the study, pregnant women were excluded in our study.

Aslo individuals using self medications were excluded, why?.

Authors: Patients who self-medicate do not engage in health-care consultations, which may obstruct our goal of determining the way to seek medical attention. Patients who self-medicate were excluded from the study to avoid the complexities of finding practice of medical care for a chronic illness.

Reviewer #2: 

Attached comments for the author:

Minor change in the tiltle " illnesses".

Authors: Many thanks. Changed it to illnesses. 

Minor comments in abstract.

Results : should be explain with more clear figure.

Authors: There were two graphs shown in the manuscript and two more figures were presented in the supplement. The flow chart of sample selection is shown in the first figure, and the Barplot of reasons for CAM use is shown in the second figure. In the manuscript, both figures were explained and clearly presented. Besides, the supplementary figures were explained in the manuscript. 

The type of referencing should be consistent.

Authors: The references are verified with Pubmed suggested AMA style.

---

## [Editor Report · Decision Letter 1]

20 Dec 2021

Prevalence and determinants of using complementary and alternative medicine for the treatment of chronic illnesses: A multicenter study in Bangladesh

PONE-D-21-26170R1

Dear Dr. Hossain,

We’re pleased to inform you that your manuscript has been judged scientifically suitable for publication and will be formally accepted for publication once it meets all outstanding technical requirements.

Kind regards,

Bashir Alsiddig Yousef, Ph.D

Academic Editor

PLOS ONE
---

## [Editor Report · Acceptance letter]

27 Dec 2021

PONE-D-21-26170R1 

Prevalence and determinants of using complementary and alternative
medicine for the treatment of chronic illnesses: A multicenter study in Bangladesh 

Dear Dr. Hossain:

I'm pleased to inform you that your manuscript has been deemed suitable for publication in PLOS ONE. Congratulations! Your manuscript is now with our production department. 

Kind regards, 

on behalf of

Dr. Bashir Alsiddig Yousef 

Academic Editor

PLOS ONE